# Longitudinal and regional association between dietary factors and prevalence of Crohn's disease in Japan

Makoto Kodama [1,2,3]*, Soh Okano[4,5], Shuko Nojri[6], Keiko Abe[1], Masayuki Fukata[5], Yoshihiro Nagase[1], Hiroko Kodama[7]

1 Department of Pathology, Tokyo Yamate Medical Center, Tokyo, Japan, 2 Department of Human Pathology, Graduate School of Medical and Dental Sciences, Tokyo Medical and Dental University, Tokyo, Japan, 3 Genomics Unit, Keio Cancer Center, Keio University School of Medicine, Tokyo, Japan, 4 Department of Human Pathology, Juntendo University School of Medicine, Tokyo, Japan, 5 Center for Inflammatory Bowel Disease, Division of Gastroenterology, Department of Internal Medicine, Tokyo Yamate Medical Center, Tokyo, Japan, 6 Medical Technology Innovation Center, Juntendo University, Tokyo, Japan, 7 Graduate School of Health Sciences, Teikyo Heisei University, Ichihara, Japan

* mkodpth1@tmd.ac.jp

**Data Availability Statement:** All relevant data are within the manuscript and its Supporting information files.

## Abstract

Although a Western diet has been identified as a risk factor for Crohn's disease (CD), there is still controversy surrounding the specific foods that may contribute to the development of the disease. In this study, we examined the association between food intake and the prevalence of CD in Japan, as Japanese patients with CD are known to have limited genetic involvement. We identified changes in food intake associated with an increase in the number of patients with CD by analyzing the per capita consumption of food types from 1981 to 2014. Additionally, we examined the association between CD prevalence and food intake in each prefecture. Finally, the relationship between food intake and estimated age at CD onset was also investigated. Between 1981 and 2014, we observed Increased consumption of meat, eggs, milk and dairy products, oil, and potatoes and decreased consumption of grains, beans, vegetables, fruit, fish, sugar, and seaweed. The annual incidence of CD increased by 1388% over the same period. We found that meat consumption was significantly associated with CD prevalence ($\beta = 0.503$, $p = 0.0003$), while a significant negative correlation was observed between CD prevalence and fruit and vegetable consumption (fruit, $\beta = 0.464$, $p = 0.0012$; vegetables, $\beta = 0.404$, $p = 0.0023$). Furthermore, we estimated that the peak consumption of more meat and less fruit and vegetables and the peak age of CD onset occurred within the age range of 20–24 years. Our study identified a clear correlation between the consumption of meat, fruits, and vegetables and the prevalence of CD in Japan. Additionally, we found an association between meat, fruit, and vegetable consumption and the age at CD onset.

## Introduction

Many environmental factors can contribute to the development of various diseases; however, establishing a causal relationship between diseases and environmental risks is challenging.

**Funding:** The author(s) received no specific funding for this work.

**Competing interests:** The authors have declared that no competing interests exist.

Crohn's disease (CD) is an inflammatory bowel disease (IBD) that causes chronic relapsing-remitting intestinal inflammation. The prevalence of CD has increased globally over the last few decades and has become a significant health and socioeconomic burden in many countries. Although the exact etiology of CD remains unknown, it is often associated with environmental, genetic, and immunologic factors [1]. A Westernized lifestyle has been identified as a major environmental factor contributing to the growing incidence of CD, as the initial increase in CD incidence occurred in Western countries [2]. While the number of patients with CD has plateaued in most Western countries, a growing incidence of CD is observed in countries undergoing Westernization, such as countries in Asia and other developing countries [3]. Several genome-wide association studies (GWAS) have revealed CD genetic risk factors. More than 200 susceptibility loci for IBD have been described, with over 70 of them related to CD development [1, 3]. Interestingly, the contribution of genetic factors differs by race, and most Asian populations have very limited associations with genetic variants in CD [4], indicating a greater involvement of environmental factors in Asian populations. Therefore, it is crucial to examine environmental risk factors for CD in Asian populations.

Japanese society has undergone rapid westernization, leading to changes in the types of diseases that have emerged in Japan over the past 70 years. In the immediate aftermath of World War II, the Japanese diet primarily consisted of traditional Japanese meals featuring rice as a staple food alongside fish and vegetables with fermented soybean seasoning. However, during the period of rapid economic growth between 1954 and 1973, the intake of animal meat, oil, fresh fruits, and dairy products increased. Even after a period of rapid economic growth, the trend of westernization was continued into the 1980s, and the number of patients with lifestyle-related diseases, such as obesity, diabetes, and hyperlipidemia, began to increase, becoming a socioeconomic and health issue [5]. In Japan, CD registration began in 1976, and the number of the patients with CD increased from 128 in 1976 to 47,633 in 2020 [6]. The incidence of CD has sharply risen since the 1980s, with environmental factors and changes in diet thought to be primarily responsible [7].

It has long been hypothesized that certain dietary components may trigger CD. Several foods have been reported as risk factors for CD, but there is still dispute over which exact foods or nutrients are associated with the development of CD [8–11]. Supporting this hypothesis has been challenging, given the diversity of populations and food consumption. However, Japan's unique situation and relatively homogenous society enable longitudinal and regional correlation studies, which are challenging to conduct in other countries. In this study, we assessed the correlation between CD prevalence and food consumption in each prefecture of Japan and found a regional correlation between CD prevalence and meat, fruit, and vegetable consumption. Our data will hopefully encourage further research on the environmental risk factors for CD development, which may lead to a reduction in the global incidence of CD.

## Materials and methods

### Study design

We identified the dietary changes associated with westernization in Japan and compared them with the incidence of CD. We conducted an ecological study to examine the effects these dietary changes had on the development of CD in Japan, where genetic involvement in the pathogenesis of CD is rare. The study design was shown in S1 Fig.

### Data collection

Data on per capita consumption of different food types from 1981 to 2014 was collected from the Food Balance Sheet of the Ministry of Agriculture of Japan (https://www.maff.go.jp/j/

zyukyu/fbs/ [Japanese], S1 Table). Food consumption data is calculated using domestic production, trade volume, and inventory. Data on 15 food types are available on the Food Balance Sheet. We excluded soy sauce and miso, which are traditional Japanese seasonings, since no association between specific seasonings and the onset of CD has been reported, and traditional Japanese seasonings are not associated with Western diets. We also excluded any starches which were classified as ingredients of processed foods. We collected data from the following 12 food types: grains, beans, vegetables, fruits, meat (both red and poultry meat), eggs, milk, dairy products, fish, sugar, oils, potatoes, and seaweed.

The Japanese national registry of CD was established in 1976 and used the diagnostic criteria for CD to provide patients with a national insurance system [12, 13]. We collected data on the annual number of patients newly diagnosed with CD in Japan from 1981 to 2014 and the number of CD patients in every prefecture as of 2014 from a report on the Public Health Administration and Services in Japan (https://www.nanbyou.or.jp/entry/5354 [Japanese], S2 Table). The population of each prefecture was obtained from the Statistical Handbook of Japan 2014 (https://www.stat.go.jp/english/data/handbook/pdf/2018all.pdf), and the prevalence of CD in each prefecture was calculated using these values.

Data on the household consumption of meat, milk, and eggs from 1981 to 2014 in each prefecture was collected from two sources. Data from 2005 to 2014 was obtained from the Family Income and Expenditure Survey conducted by the Statistics Bureau of the Ministry of Internal Affairs and Communications, which is publicly available online (https://www.stat.go.jp/data/kakei/rank/singleyear.html[Japanese]). Data from 1981 to 2004 was taken from the National Diet Library of Japan, which collects and stores all publications in Japan. This data was collected from approximately 9,000 randomly selected individuals in Japan. We calculated the average per-household meat, milk, and egg consumption in each prefecture from 1981 to 2014. However, although the consumption of dairy products increased in this timeframe, we did not collect data on dairy product consumption in each prefecture. This is because consumption data for dairy products is calculated by price, not quantity, and there are too many variables for dairy products per region, making it challenging to accurately calculate consumption. Furthermore, the method of data collection differs between the Family Income and Expenditure Survey and the Food Balance Sheet, which calculates data on dairy product consumption using milk quantity.

The estimated age of CD onset was assessed using data from the Research Committee of Inflammatory Bowel Disease and the Ministry of Health and Welfare of Japan [6]. The average meat, vegetable, and fruit consumption in each age group was collected from the National Health and Nutrition Survey by the Ministry of Health, Labor, and Welfare.

### Statistical analysis

Linear regression analysis was used to assess the correlation between food type and CD prevalence in each prefecture and was visualized using GraphPad Prism 7 (GraphPad Software, San Diego, CA, USA). Statistical significance was set at $P < 0.05$.

### Results

Initially, we assessed the longitudinal changes in food consumption per item in Japan. We observed a rise in the consumption of meat, eggs, milk and dairy products, oil, and potatoes and a decline in the consumption of grains, beans, vegetables, fruit, fish, sugar, and seaweed between 1981 and 2014 (Fig 1A and 1B). This data exemplifies the dietary westernization that occurred in Japan during this period. The annual incidence of CD increased over the same period, demonstrating a 1388% increase in 2014 as compared to 1981 (Fig 1A).

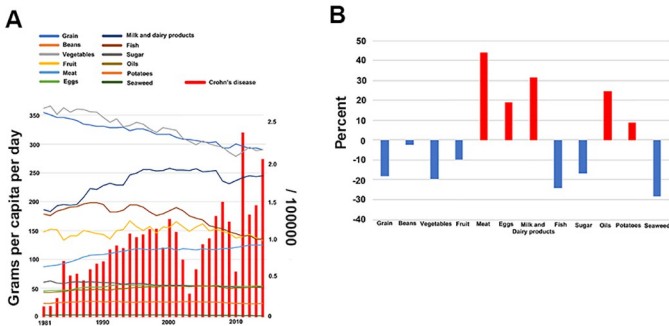

**Fig 1.** (A). Graph showing the transition in the number of patients with Crohn's disease and the transition in food consumption from 1981 to 2014. (B). Percentage change in food consumption between 1981 and 2014.

Increased consumption of animal meat, eggs, fat (or oil), sugar, and fish has been associated with an increased risk of developing CD [14–20]. Conversely, fruits and vegetables have been believed to lower the risk of CD development [14, 17, 18]. To examine the relationship between these food items and the development of CD, we compared the regional consumption of these food items and the prevalence of CD across Japan's 47 prefectures. Among all food items, meat consumption was the only one that demonstrated a significant positive correlation with CD prevalence (meat, β = 0.503, p = 0.0003; Fig 2A–2H, Table 1). The heat maps for CD prevalence and daily meat consumption per household were found to be matching. However, a few prefectures demonstrated a relative mismatch (Fig 2I and 2J). It is possible that certain

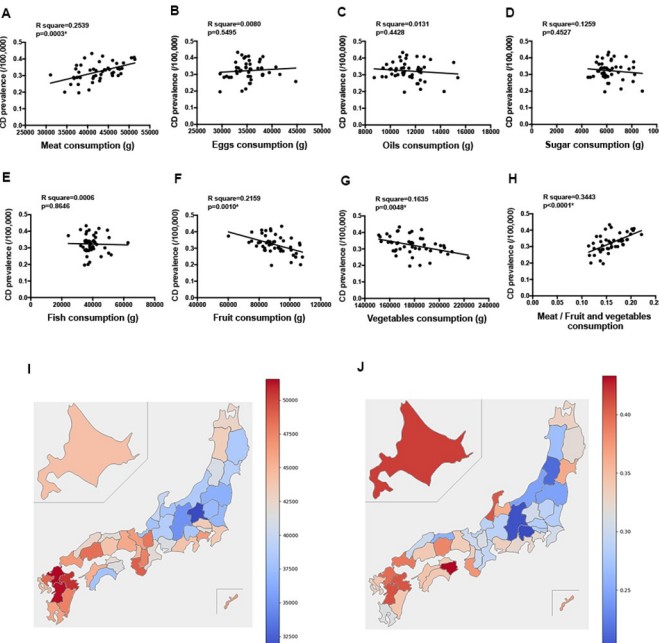

**Fig 2.** Correlation between the prevalence of patients with Crohn's disease and meat consumption (A), egg consumption (B), Oil consumption (C), Sugar consumption (D), fish consumption (E), fruit consumption (F), and vegetable consumption (G) comparing each prefecture. (H). Correlation between the prevalence of patients with Crohn's disease and meat consumption divided by vegetable and fruit consumption. (I). Heat map of meat consumption in each prefecture (J). Heat map of the prevalence of patients with Crohn's disease in each prefecture.

**Table 1. Regression table between variables and CD prevalence.**

|  | β | R² | F | *p* value | 95% CI |
|---|---|---|---|---|---|
| Meat | 0.503 | 0.2539 | 15.32 | 0.0003 | 2.9e-006 to 9.0e-006 |
| Eggs | 0.089 | 0.080 | 0.3637 | n.s. | -3.8e-006 to 7.1e-006 |
| Oils | 0.114 | 0.013 | 0.5996 | n.s. | -1.6e-005 to 7.3e-006 |
| Sugar | 0.109 | 0.0125 | 0.5739 | n.s. | -2.3e-005 to 1.0e-005 |
| Fish | 0.024 | 0.0006 | 0.0294 | n.s. | 12.8e-006 to 2.4e-006 |
| Fruit | 0.4646 | 0.2159 | 12.39 | 0.0010 | -4.0e-006 to -1.0e-006 |
| Vegetables | 0.4043 | 0.1635 | 8.799 | 0.0048 | -2.2e-006 to -4.33-007 |
| Meat / Fruit and vegetables | 0.5797 | 0.3361 | 22.78 | <0.0001 | 0.7728 to 1.901 |

inconsistencies between meat consumption and CD prevalence can be attributed to the consumption of vegetables and fruit. In contrast to meat consumption, a significant negative correlation was observed between CD prevalence and the consumption of fruits and vegetables (fruit, β = 0.464, p = 0.0012; Fig 2F; vegetables, β = 0.404, p = 0.0023; Fig 2G). When we compared CD prevalence and the ratio of meat consumption to fruit or vegetable consumption, a higher correlation was observed than in their individual comparisons (β = 0.586, p<0.0001; Fig 2H).

While patients with CD are typically diagnosed between the ages of 15 and 30, the scientific basis of diagnosis in this age group has not yet been specifically determined. When we examined age-based trends in food consumption, we observed a peak in meat consumption between the ages of 15–19 years old for both males and females in Japan (Fig 3A). Additionally, a decrease in the consumption of fruits and vegetables was observed in individuals aged between 20–30 and 20 years, respectively (Fig 3B and 3C). The peak age for the consumption of more meat and fewer fruits and vegetables, along with the peak age of CD onset, was estimated to be in a relatively younger generation, aged 20–24 (Fig 3D). These trends are consistent with the association between the consumption of meat, fruits, and vegetables and the prevalence of CD in Japan.

## Discussion

CD affects approximately 300 out of 100,000 people in Europe and North America, and its global prevalence has been increasing, posing a socioeconomic burden [1]. The exact cause of

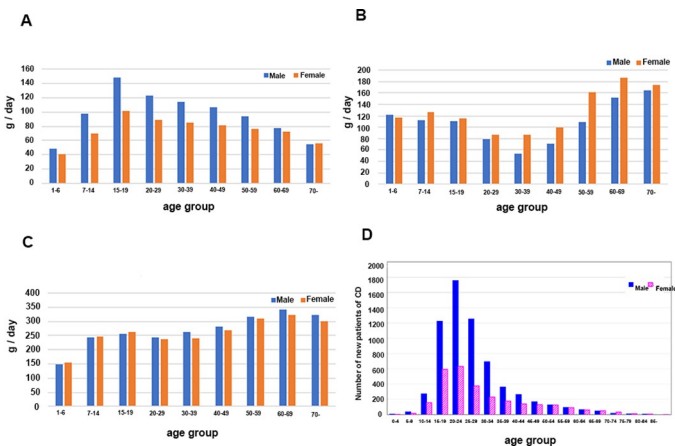

**Fig 3.** (A). Meat consumption by age group. (B). Vegetable consumption by age group (C). Fruit consumption by age group (D). Age of Crohn's disease onset.

CD remains unknown, but environmental factors are believed to trigger the disease's development in genetically susceptible individuals by causing mucosal immune abnormalities [3]. Through identifying the exact triggers, we may be able to prevent the development of CD. Although several studies have highlighted potential environmental triggers, it is still unclear whether any specific foods or nutrients can cause CD [8–11]. This is likely due to the challenges of conducting dietary studies in diverse populations. In this study, we aimed to identify longitudinal and regional correlations between food consumption and the prevalence of CD in Japan, where significant changes in dietary habits and CD incidence have been observed over the past few decades. Our results indicate a longitudinal correlation between the increased incidence of CD and the rising consumption of meat, along with reduced intake of fruits and vegetables. This pattern of food consumption was positively correlated with CD prevalence and, in some cases, may be responsible for triggering the onset of the disease.

Foods are believed to be among the most significant environmental factors in the development and relapse of CD, a disease characterized by inflammation of the gastrointestinal tract [18]. Previous studies have suggested that a high animal protein intake is linked to a higher prevalence of CD in Western countries, as opposed to Asian and other developing countries [20, 25]. However, this association was not observed in some recent prospective cohort studies in North America and Europe 9,10. The difficulty in detecting environmental risk factors in Western countries might be due to the identification of more than 70 susceptible gene mutations for CD development in Western populations [3]. In contrast, only a limited number of genetic variants associated with CD have been found in most Asian populations, including the Japanese [4]. As an island nation, Japan has been a relatively homogenous society, and the lifestyle and dietary habits of the Japanese tend to be uniform [21]. These unique features of the Japanese population are advantageous when examining environmental factors involving CD development.

Three previous reports have examined the correlation between food consumption and CD prevalence or development in Japan. Of these reports, two were epidemiological studies [14, 15], and one was a case-control study [16]. Our findings were consistent with the results of epidemiological studies but not with those of the case-control study. While the epidemiological study found an association between meat consumption and CD prevalence, the case-control study found no correlation between the two. It is important to interpret and consider the discrepancies in these results carefully. In regards to IBD, a case-control study has been known to cause errors in determining the risk factors, as the pre-diagnosed period prior to overt symptoms can extend several years, and individuals occasionally change their diet during that period [22]. Our findings may have implications for the prevention and management of CD in other countries, particularly in industrialized developing countries where a rapid rise in patients with CD [1], along with increased meat consumption, has been observed [23]. In Japan, meat consumption continues to rise alongside Westernization, and so does the number of patients with CD. Although it is difficult to establish a direct causal relationship, we may need to reevaluate our dietary habits.

Although CD tends to occur in young adults, the environmental factors associated with its age-related incidence have yet to be explored. Our study found that the young adult population in Japan had increased their meat consumption while decreasing their consumption of fruits and vegetables. Based on our results and other studies [24, 25], it is reasonable to conclude that an immoderate imbalance in meat, fruit, and vegetable consumption could trigger the development of CD.

However, this study has several limitations. Since part of this study relied on self-reported data, recall bias might occur. Data regarding food consumption were not obtained directly from patients with CD. Therefore, we were unable to precisely determine their actual food

intake. Further examination at the individual level is needed because the present study may in the ecological fallacy. In addition, although this study was limited to food, analysis including other environmental factors is also needed.

## Conclusions

By comparing food consumption and the distribution of patients with CD by prefecture over several decades, we found a clear correlation between the consumption of meat, fruits, and vegetables and the prevalence of CD in Japan. We also demonstrated an association between the age at CD onset and meat, fruit, and vegetable consumption. As Japanese patients with CD have limited genetic involvement, environmental factors may be more pronounced in the pathogenesis of CD.

## Supporting information

**S1 Fig. The study design.** The study design of this research.
(TIF)

**S1 Table. The average consumptions of each food from 1980 to 2014.** The study design of this research.
(XLSX)

**S2 Table. Prevalence of the patients with CD.** The prevalence of the patients with CD in each prefecture and in each year (1981–2014).
(XLSX)

## Author Contributions

**Conceptualization:** Makoto Kodama.

**Data curation:** Makoto Kodama, Shuko Nojri.

**Investigation:** Makoto Kodama, Soh Okano.

**Supervision:** Keiko Abe, Hiroko Kodama.

**Visualization:** Makoto Kodama, Yoshihiro Nagase.

**Writing – original draft:** Makoto Kodama.

**Writing – review & editing:** Masayuki Fukata.

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
