## [Decision Letter · Decision Letter 0]

17 Jan 2024

PONE-D-23-25250Longitudinal and regional association between dietary factors and prevalence of Crohn’s disease in JapanPLOS ONE

Dear Dr. Kodama,

Thank you for submitting your manuscript to PLOS ONE. After careful consideration, we feel that it has merit but does not fully meet PLOS ONE’s publication criteria as it currently stands. Therefore, we invite you to submit a revised version of the manuscript that addresses the points raised during the review process.

We look forward to receiving your revised manuscript.

Kind regards,

Peivand Bastani

Academic Editor

PLOS ONE

Journal Requirements:

2. We note that your Data Availability Statement is currently as follows: “All relevant data are within the manuscript and its Supporting Information files.”

3. We note that Figures 2 (I, J) in your submission contain copyrighted images. All PLOS content is published under the Creative Commons Attribution License (CC BY 4.0), which means that the manuscript, images, and Supporting Information files will be freely available online, and any third party is permitted to access, download, copy, distribute, and use these materials in any way, even commercially, with proper attribution. For more information, see our copyright guidelines: http://journals.plos.org/plosone/s/licenses-and-copyright.

     a. You may seek permission from the original copyright holder of Figures 2 (I, J) to publish the content specifically under the CC BY 4.0 license.

Reviewers' comments:

Reviewer's Responses to Questions

**Comments to the Author**

1. Is the manuscript technically sound, and do the data support the conclusions?

Reviewer #1: Yes

Reviewer #2: Yes

2. Has the statistical analysis been performed appropriately and rigorously? 

Reviewer #1: I Don't Know

Reviewer #2: Yes

3. Have the authors made all data underlying the findings in their manuscript fully available?

Reviewer #1: No

Reviewer #2: Yes

4. Is the manuscript presented in an intelligible fashion and written in standard English?

Reviewer #1: Yes

Reviewer #2: Yes

5. Review Comments to the Author

Reviewer #1: Dear authors,

Your study is a valuable research with an interesting topic that aims to investigate the longitudinal and regional association between dietary factors and prevalence of Crohn’s disease in Japan. To improve the study, it is better to consider the following comments:

1- Make the introduction shorter.

2- Mention a statistic of Crohn's disease in Japan in the introduction.

3- The design of study is not mentioned.

4- Write about the ecological fallacy of your research in the limitation section.

5- I did not see the regression table in the results.

6- The effect measure for correlations should be written in the results.

7- please provide ethical code of your research in Method section.

Reviewer #2: General comment:

The topic approached by authors of this paper is interesting and overall, I believe the study has relevant information hereby suitable for publication after revision.

The authors clearly explained their work throughout the manuscript. I just have some remarks/suggestions to point out.

The study found a correlation between increased meat consumption and CD prevalence in Japan, as well as reduced intake of fruits and vegetables. Moreover, it found a peak in meat consumption between the ages of 15-19 years old for both males and females in Japan, as well as a decrease in the consumption of fruits and vegetables in individuals aged between 20-30 and 20 years, respectively.

Specific comments:

In Abstract:

- The results should be reported in a statistical and accurate manner. (for example: increase in meat consumption: which percent or value? The P-value?)

In Methods:

- Meat consumption: the author mean just red meat or is poultry meat included?

In Results & Discussion:

- Fig 3: The units in the horizontal and vertical axes are not specified. Therefore, this figure cannot be understood on its own.

- The study relied on self-reported data, which may be subject to recall bias and other inaccuracies.

- The study did not take into account other potential environmental factors such as smoking, air pollution, and other lifestyle factors.

6. PLOS authors have the option to publish the peer review history of their article (what does this mean?). If published, this will include your full peer review and any attached files.

Reviewer #1: No

Reviewer #2: No

---

## [Author Response · Author response to Decision Letter 0]

29 Jan 2024

Dear Editor, 

We were pleased to have an opportunity to revise our manuscript entitled “Longitudinal and regional association between dietary factors and prevalence of Crohn’s disease in Japan”. In revised manuscript, we have carefully considered reviewers’ comments and suggestions. As instructed, we have attempted to succinctly explain changes made in reaction to all comments. We reply to each comment in point-by-point fashion. The revised manuscript was highlighted in red. 

Editor suggestions

●Response: We checked templates and modified the style as indicated.

(p2 line29; p4 line65, p8 line124,125,131,134; p10 line177; p11 line184; p15 line 245; p19 line311, 320; p20 line326)

2. We note that your Data Availability Statement is currently as follows: “All relevant data are within the manuscript and its Supporting Information files.” Please confirm at this time whether or not your submission contains all raw data required to replicate the results of your study. Authors must share the “minimal data set” for their submission. PLOS defines the minimal data set to consist of the data required to replicate all study findings reported in the article, as well as related metadata and methods (https://journals.plos.org/plosone/s/data-availability#loc-minimal-data-set-definition). For example, authors should submit the following data: - The values behind the means, standard deviations and other measures reported; - The values used to build graphs; - The points extracted from images for analysis. Authors do not need to submit their entire data set if only a portion of the data was used in the reported study. If your submission does not contain these data, please either upload them as Supporting Information files or deposit them to a stable, public repository and provide us with the relevant URLs, DOIs, or accession numbers. For a list of recommended repositories, please see https://journals.plos.org/plosone/s/recommended-repositories. If there are ethical or legal restrictions on sharing a de-identified data set, please explain them in detail (e.g., data contain potentially sensitive information, data are owned by a third-party organization, etc.) and who has imposed them (e.g., an ethics committee). Please also provide contact information for a data access committee, ethics committee, or other institutional body to which data requests may be sent. If data are owned by a third party, please indicate how others may request data access. 

●Response: We have added sets of the data relating to this article as supporting information files (S1 table and S2 table).

3. We note that Figures 2 (I, J) in your submission contain copyrighted images. All PLOS content is published under the Creative Commons Attribution License (CC BY 4.0), which means that the manuscript, images, and Supporting Information files will be freely available online, and any third party is permitted to access, download, copy, distribute, and use these materials in any way, even commercially, with proper attribution. For more information, see our copyright guidelines: http://journals.plos.org/plosone/s/licenses-and-copyright. We require you to either (1) present written permission from the copyright holder to publish these figures specifically under the CC BY 4.0 license, or (2) remove the figures from your submission: a. You may seek permission from the original copyright holder of Figures 2 (I, J) to publish the content specifically under the CC BY 4.0 license. We recommend that you contact the original copyright holder with the Content Permission Form (http://journals.plos.org/plosone/s/file?id=7c09/content-permission-form.pdf) and the following text: “I request permission for the open-access journal PLOS ONE to publish XXX under the Creative Commons Attribution License (CCAL) CC BY 4.0 (http://creativecommons.org/licenses/by/4.0/). Please be aware that this license allows unrestricted use and distribution, even commercially, by third parties. Please reply and provide explicit written permission to publish XXX under a CC BY license and complete the attached form.” Please upload the completed Content Permission Form or other proof of granted permissions as an "Other" file with your submission. In the figure caption of the copyrighted figure, please include the following text: “Reprinted from [ref] under a CC BY license, with permission from [name of publisher], original copyright [original copyright year].” b. If you are unable to obtain permission from the original copyright holder to publish these figures under the CC BY 4.0 license or if the copyright holder’s requirements are incompatible with the CC BY 4.0 license, please either i) remove the figure or ii) supply a replacement figure that complies with the CC BY 4.0 license. Please check copyright information on all replacement figures and update the figure caption with source information. If applicable, please specify in the figure caption text when a figure is similar but not identical to the original image and is therefore for illustrative purposes only. 

●Response: We used statistical software “Tableau” to create the figure 2(I, J). There are free and paid versions of "Tableau". We used a free version and we think that’s because these figures became copyrighted images. The data and figures were our original so we re-created these figures using the software, ”Python”. 

●Response: We reviewed references and confirmed that it is complete and correct. There is no retracted article in our references.

Reviewer #1 comments

1- Make the introduction shorter. 

●Response: We removed several sentences in the introduction.

(p4, line72,73; p4,5 line95-101; p6 line110)

2- Mention a statistic of Crohn's disease in Japan in the introduction. 

●Response: We have added a statistic of Crohn’s disease in Japan in the introduction.

(p6 line106,107)

3- The design of study is not mentioned. 

●Response: We created the figure (S1 Fig) about the design of study and mentioned about it in the first paragraph of materials and methods section. 

(p8 line130)

4- Write about the ecological fallacy of your research in the limitation section. 

●Response: We added about the ecological fallacy of my research in the limitation section.

 (p18,19 line306-309)

5- I did not see the regression table in the results. 

●Response: We created the regression table in the results.

(Table 1.)

6- The effect measure for correlations should be written in the results. 

●Response: We added the effect measure (Standardized regression coefficient) in the results.

(p12 line204,212; p13 line215)

7- please provide ethical code of your research in Method section. 

●Response: We used only the existing public data, therefore we did not have ethical code for our research.

Reviewer #2 comments

In Abstract: 

- The results should be reported in a statistical and accurate manner. (for example: increase in meat consumption: which percent or value? The P-value?) 

●Response: We added the statistical value in the abstract.

(p2 line44,46) 

In Methods: 

- Meat consumption: the author mean just red meat or is poultry meat included? 

●Response: Since we Japanese generally do not distinguish between red meat and poultry (white) meat, "Meat" means both red meat and white meat, and we added that discription.

(p9 line142) 

In Results & Discussion: 

- Fig 3: The units in the horizontal and vertical axes are not specified. Therefore, this figure cannot be understood on its own. 

●Response: We put the units in the horizontal and vertical axes of Fig 3.

- The study relied on self-reported data, which may be subject to recall bias and other inaccuracies.

●Response: We added about recall bias and other inaccuracies in the discussion section.

(p18,19 line304-305,306-309)

 - The study did not take into account other potential environmental factors such as smoking, air pollution, and other lifestyle factors.

●Response: We mentioned about other environmental factors in the limitation section.

(p18,19 line306-309)

CONCLUDING REMARKS: Again, thank you for giving us the opportunity to strengthen our manuscript with your valuable comments and queries. We have worked hard to incorporate your feedback and hope that these revisions persuade you to accept our submission. As we revise the article, we re-organized references, and we added another co-author, Yoshihiro Nagase.

---

## [Decision Letter · Decision Letter 1]

1 Mar 2024

Longitudinal and regional association between dietary factors and prevalence of Crohn’s disease in Japan

PONE-D-23-25250R1

Dear Dr. Kodama,

We’re pleased to inform you that your manuscript has been judged scientifically suitable for publication and will be formally accepted for publication once it meets all outstanding technical requirements.

Kind regards,

Peivand Bastani

Academic Editor

PLOS ONE

Additional Editor Comments (optional):

Reviewers' comments:

Reviewer's Responses to Questions

**Comments to the Author**

1. If the authors have adequately addressed your comments raised in a previous round of review and you feel that this manuscript is now acceptable for publication, you may indicate that here to bypass the “Comments to the Author” section, enter your conflict of interest statement in the “Confidential to Editor” section, and submit your "Accept" recommendation.

Reviewer #1: All comments have been addressed

Reviewer #2: All comments have been addressed

Reviewer #3: All comments have been addressed

2. Is the manuscript technically sound, and do the data support the conclusions?

Reviewer #1: Yes

Reviewer #2: Yes

Reviewer #3: Yes

3. Has the statistical analysis been performed appropriately and rigorously? 

Reviewer #1: Yes

Reviewer #2: Yes

Reviewer #3: I Don't Know

4. Have the authors made all data underlying the findings in their manuscript fully available?

Reviewer #1: Yes

Reviewer #2: Yes

Reviewer #3: Yes

5. Is the manuscript presented in an intelligible fashion and written in standard English?

Reviewer #1: Yes

Reviewer #2: Yes

Reviewer #3: Yes

6. Review Comments to the Author

Reviewer #1: Dear authors,

First, I thank you for the answers you gave to the reviewers, but I need to say some points:

1- The correlation index between two quantitative variables is r, and if the relationship between the two variables is inverse, the r becomes negative, but what you reported is beta. Please check again.

2- Every research project is approved in the relevant university and gets a code. Have you approved your project in the university?

Reviewer #2: All comments have been addressed. In my opinion, there are no scientific problems with the article. It can be printed.

Reviewer #3: (No Response)

7. PLOS authors have the option to publish the peer review history of their article (what does this mean?). If published, this will include your full peer review and any attached files.

Reviewer #1: No

Reviewer #2: No

Reviewer #3: No

---

## [Editor Report · Acceptance letter]

29 Apr 2024

PONE-D-23-25250R1 

PLOS ONE

Dear Dr. Kodama, 

I'm pleased to inform you that your manuscript has been deemed suitable for publication in PLOS ONE. Congratulations! Your manuscript is now being handed over to our production team.

Kind regards, 

on behalf of

Dr Peivand Bastani 

Academic Editor

PLOS ONE